# Sink Strength Promoting Remobilization of Non-Structural Carbohydrates by Activating Sugar Signaling in Rice Stem during Grain Filling

**DOI:** 10.3390/ijms23094864

**Published:** 2022-04-27

**Authors:** Zhengrong Jiang, Qiuli Chen, Lin Chen, Dun Liu, Hongyi Yang, Congshan Xu, Jinzhi Hong, Jiaqi Li, Yanfeng Ding, Soulaiman Sakr, Zhenghui Liu, Yu Jiang, Ganghua Li

**Affiliations:** 1College of Agronomy, Nanjing Agricultural University, Nanjing 210095, China; zhengrong.jiang@inrae.fr (Z.J.); 2019101047@njau.edu.cn (Q.C.); linchen@njau.edu.cn (L.C.); 2020101041@stu.njau.edu.cn (D.L.); 2021101032@stu.njau.edu.cn (H.Y.); 2018201012@njau.edu.cn (C.X.); 11120223@stu.njau.edu.cn (J.H.); 2021101031@stu.njau.edu.cn (J.L.); dingyf@njau.edu.cn (Y.D.); liuzh@njau.edu.cn (Z.L.); yujiang@njau.edu.cn (Y.J.); 2Institut Agro, University of Angers INRAE, IRHS, SFR QUASAV, F-49000 Angers, France; soulaiman.sakr@agrocampus-ouest.fr

**Keywords:** rice, sink strength, grain filling, non-structural carbohydrates, remobilization, sugar

## Abstract

The remobilization of non-structural carbohydrates (NSCs) in the stem is essential for rice grain filling so as to improve grain yield. We conducted a two-year field experiment to deeply investigate their relationship. Two large-panicle rice varieties with similar spikelet size, CJ03 and W1844, were used to conduct two treatments (removing-spikelet group and control group). Compared to CJ03, W1844 had higher 1000-grain weight, especially for the grain growth of inferior spikelets (IS) after removing the spikelet. These results were mainly ascribed to the stronger sink strength of W1844 than that of CJ03 contrasting in the same group. The remobilization efficiency of NSC in the stem decreased significantly after removing the spikelet for both CJ03 and W1844, and the level of sugar signaling in the T6P-SnRK1 pathway was also significantly changed. However, W1844 outperformed CJ03 in terms of the efficiency of carbon reserve remobilization under the same treatments. More precisely, there was a significant difference during the early grain-filling stage in terms of the conversion of sucrose and starch. Interestingly, the sugar signaling of the T6P and SnRK1 pathways also represented an obvious change. Hence, sugar signaling may be promoted by sink strength to remobilize the NSCs of the rice stem during grain filling to further advance crop yield.

## 1. Introduction

Rice (*Oryza sativa* L.) is one of the most important crops for the burgeoning world population [1]. Due to the high demand for food, the cultivation of large-panicle rice, exhibiting high sink capacity with numerous spikelets per panicle, has been widely concerned about enhancing yield. The grain filling period is a vital period for the supply of carbohydrates (source) and the capacity of grains to utilize carbohydrates (sink) for regulation [2,3,4]. Nevertheless, the grain-filling ability (sink strength), affected by sink capacity and sink activity, is different in distinct varieties of large-panicle rice [5]. The inferior spikelet (IS), located at the basal secondary branches, has poor grain filling, inducing a limitation of the improvement in sink strength [6,7]. Interestingly, the low sink strength of the IS is also associated with the poor grain filling of panicles and low yield [8,9]. Enough source supply from the stem and leaf is necessary for grain filling of the IS and the rice yield during the grain filling period [7,10]. Thus, it is essential to understand the relationship between sink strength and source ability for enhancing yield.

The stem acts as a non-photosynthetic source organ, supplying sucrose for sink growth from transforming leaf photosynthesis and non-structural carbohydrates (NSCs), which is vital in the accumulation and transport of photo-assimilates [11,12]. Indeed, during the grain filling stage, some parts of photo-assimilates are transported to the stems, stored as starch in rice, and then degraded and transported to grains for the grain-filling process [13,14]. The NSCs of stems are important to the sink strength and grain-filling efficiency, contributing 10–40% to rice yield [15,16]. The activities of sucrose synthase (EC 2.4.1.13) and ADP glucose pyrophosphorylase (EC 2.7.7.27) play important roles in the starch storage of the stem, and α-amylase (EC 3.2.1.1) is essential for the degradation of starch [12,17]. Hence, high activities of starch to sucrose conversion may be responsible for NSCs’ remobilization and transportation. However, the regulation mechanism between the sink growth and NSCs’ remobilization of rice stems is still unclear.

Photo-assimilates accumulate in rice stems through a portion of processes in the carbohydrates’ metabolism [18]. As an important signal in regulating plant development, sugar signaling has been identified as a critical determinant for carbohydrates’ metabolism in almost all aspects of plant development, but little is known about the regulation of sugar signaling to assimilate storage and remobilization in the stems of rice [19]. As a signal of sugar availability, trehalose-6-phosphate (T6P) is synthesized from UDP-glucose and glucose 6-phosphate by trehalose 6-phosphate synthase (TPS), and dephosphorylated to trehalose by trehalose 6-phosphate phosphatase (TPP). High T6P levels are positively correlated with high sucrose contents and high metabolic status [20]. The low T6P content of sinks can act as a starvation signal to upregulate sink strength when improving the sucrose movement to sinks [21]. Intriguingly, the sugar signaling of Snf1-related kinase 1 (SnRK1) is particularly activated by sugar deprivation in regulating carbon allocation and utilization, inhibited by T6P signaling [22,23]. The pathway of T6P-SnRK1 signaling can respond to sucrose induced by priming growth recovery [24]. Therefore, the balance of sink demand and NSCs in stems derived from sucrose transportation may be critically determined by the regulation of sugar signaling in stems.

As reviewed above, the physiological role of NSCs in stems has been investigated by many studies [13,15,25]. However, fewer studies have declared the causation between the sink strength of the panicle and the remobilization of NSCs in stems, especially for the grain-filling period. In this paper, we try to fill this gap by conducting a two-year field experiment where the two large-panicle rice cultivars (CJ03 and W1844) had two-thirds of panicles removed and none removed, respectively. The insight into the deep relationship between the sink strength and the NSCs in stems during the grain-filling stage is investigated to provide a new basis for improving rice yield and quality. In combination with grain growth and the remobilization of NSCs in the stem, we aim to build a better rice agronomic management and cropping system.

## 2. Results

### 2.1. Grain Weight and Seed Setting Rate

As shown in Table 1, the spikelet number of panicles was similar between CJ03 and W1844 in the T0 group, but the seed setting rate was not high in the field experiments of 2019 and 2020. Compared with the T0 group, the seed setting rate significantly increased by removing spikelets in CJ03 and W1844. Interestingly, the 1000-grain weight of W1844 improved considerably after spikelet thinning, while the 1000-grain weight of the T1 group in CJ03 was still significantly lower than that in the T0 group.

The noticeable changes in the 1000-grain weight and seed setting rate of the IS were quite different by thinning spikelets in CJ03 and W1844 (Table 2). The 1000-grain weight and seed setting rate of the IS in the T0 group were significantly lower than those in the SS of the two materials. After spikelet thinning, the IS of W1844 exhibited greater improvement in 1000-grain weight and seed setting rate than those in the IS of CJ03, which could be significantly improved and even higher than the level of SS in the T0 group.

### 2.2. Dynamics of DM Accumulation and NSC Content

The two-year field experiment showed a rapid increase in DM accumulation of the panicle from 8 DPA to 20 DPA, increasing slowly from 4 DPA to 8 DPA, and 20 DPA to 40 DPA (Figure 1A). Interestingly, the DM accumulation of SS in the T0 group and IS in the T1 group showed similar trends, which improved rapidly from 8 DPA to 20 DPA (Figure 1B). The grain weight of IS in W1844 improved obviously and even approached the level of SS after thinning spikelets, while the grain weight of the IS in the T1 group of CJ03 could not approach the level of SS from 8 DPA to 20 DPA (Figure 1B). Similarly, the DM accumulation of panicles in W1844 showed a higher increase than CJ03 from 8 DPA to 20 DPA (Figure 1A). This finding means that the high source ability might be needed to satisfy the high sink demand in CJ03 and W1844 from 8 DPA to 20 DPA.

The curves of the DM and NSC contents in stems showed similar trends, decreasing notably from 8 DPA to 20 DPA and significantly after removing spikelets (Figure 2). Notably, the DM and NSCs of the stems in the T1 group decreased less than those in the T0 group from 8 DPA to 20 DPA, and the remobilization rate of the T1 group was less than the T0 group (Table 3). During this period, most of the data about DM and NSCs in W1844 decreased more than those in the same treatment of CJ03, and the remobilization rate showed similar trends (Table 3). Generally, these findings suggested a far more important role in the remobilization and transportation of NSCs in the stem for grain filling of W1844 than those of CJ03 from 8 DPA to 20 DPA.

### 2.3. Metabolism of Carbohydrates in Stem

The carbohydrate content of the stem in CJ03 and W1844 during the grain-filling period was investigated. After removing the spikelets, the sucrose content and starch content were both increased in the stems of CJ03 and W1844 (Table 4). Being the key enzymes of carbohydrates’ metabolism, the SuSase and AGPase significantly increased at 8 DPA and 20 DPA after removing the spikelets (Figure 3). Additionally, the gene expression of *OsSUS2* and *OsAGPL1* showed obvious upregulation after removing the spikelets (Figure 3). Interestingly, the α-Amylase activity and gene expression of *α-Amy3* in the T1 treatment were significantly lower than those of the T0 treatment (Figure 3). Comparing the varieties of CJ03 and W1844 under the same treatment, the sugar and starch content of W1844 was not significantly higher than that of CJ03, and most of these data were significantly lower than those of CJ03 during grain filling (Table 4). At the early grain-filling stage, the gene expressions of *OsSuS2* and *OsAGPL1* in W1844 were significantly lower than those of CJ03 compared within the same treatment, and the α-Amylase activity and expression of *α-Amy3* in W1844 were lower than those of CJ03 at 8 DPA (Figure 3). Nevertheless, the trends of SuSase (*OsSuS2*) and AGPase (*OsAGPL1*) in W1844 were not significantly lower than those of CJ03 during the middle grain-filling stage, and no significant differences were detected for the trend of α-Amylase (*α-Amy3*) at 20 DPA (Figure 3). Interestingly, most of the values for CJ03 of the sucrose content and starch content were nearly equal to those from 8 DPA to 20 DPA, but a huge decrease occurred in W1844 during that period (Table 4). Generally, these data indicate that a difference in the carbohydrate metabolism in stems existed between CJ03 and W1844.

### 2.4. Regulation of Sugar Signaling in Stem

Based on the observed difference in the carbon reserve remobilization of the stems, the level of T6P and SnRK1 was possibly involved in the regulation of carbon metabolism in the stems. The T6P content and expression of *OsTPS8* in stems were notably increased in both CJ03 and W1844 after removing the spikelets, and the expression of *OsTPP1* was downregulated, respectively (Figure 4). In addition, the expression of *OSK1* and *OSK24* in the stem showed obvious downregulation after removing the spikelets (Figure 5). At 8 DPA, the T6P content and expression of *OsTPS8* in W1844 were obviously less than those of CJ03 compared within same treatment (Figure 4). Correspondingly, there were significant upregulation of T6P content and expression of *OsTPS8* occurring in the removing-spikelet treatment at the middle grain-filling stage. These sharp changes in the sugar-signaling network of the stem might be related to the difference in sink strength.

## 3. Discussion

### 3.1. Effects of Different SinK Strength to Soluble Carbohydrates’ Transportation in Rice Stem during Grain Filling

In the source–sink context, carbohydrates are produced by photosynthesis and transported to the stems of rice for carbon consumption and storage during grain filling [13]. Carbohydrates, primarily sucrose, are essential for grain growth, which might be insufficient for the grain filling of inferior spikelets (IS) during the grain filling stage [7]. To characterize the relationship between the carbohydrates of stems and grain growth in rice, we conducted two experiments in 2019 and 2020, respectively, to examine the difference in grain growth of CJ03 and W1844 after removing spikelets to force enough carbohydrates’ transport to sink. After removing spikelets, the DM weight and the NSC content of stems were significantly increased in CJ03 and W1844 during the grain filling period (Figure 2). The DM weight of the IS was also obviously improved in CJ03 and W1844 for this period (Table 2, Figure 1). Thus, we demonstrated that the carbon reserve of the stem was enough for the demand of grain growth in CJ03 and W1844 after removing spikelets. Interestingly, the grain weight and seed setting rate in CJ03 could not reach the level of the T0 group after removing spikelets, which was strictly limited by the grain growth of the IS (Table 1 and Table 2). Meanwhile, the grain growth of W1844 showed a notable improvement after removing spikelets, reaching an even higher level than that of the T0 group, especially for the grain filling of the IS (Table 1 and Table 2). According to a previous study, the grain growth of W1844 shows more efficient grain filling of the IS than that of CJ03, which is mainly governed by the strength of the sink activity [5]. In accordance with this, our previous studies on the initiation of inferior grain filling established that the poor conversion of sucrose to starch limited the grain-filling initiation of the IS [4]. Taken together, these findings indicate that low sink activity may be a direct limitation for grain filling, and the IS of W1844 shows better sink activity than that of CJ03.

Sink activity and carbohydrate transportation efficiency are both important to rice yields [12]. The phenotype of CJ03 and W1844 on grain growth is interesting because it contains large changes in the carbohydrates of the stem (Figure 1 and Figure 2). The utilization ability of carbohydrates in the sink of W1844 was better than that of CJ03 during the grain-filling period (Table 2, Figure 1 and Figure 2), and the transportation efficiency of soluble carbohydrates in the stems of W1844 was higher than that of CJ03 (Table 3, Figure 2). This operational feature of the efficiency of carbohydrates’ transportation in the stem was previously proved through studies on the initiation of inferior grain filling [5]. It kept a lower efficiency of carbohydrates’ transport in the stem in the T1 group than that of the T0 group (Table 3 and Table 4) when removal of the spikelet was conducted in panicles to decrease the sink strength in our study (Table 1, Figure 1). The increase in sink strength can promote the phloem loading in sugarcane [26]. Moreover, the poor unloading of sink is a possible limitation for NSC transportation in the stems of rice [25]. Hence, sink strength is a key determinant for the transportation of soluble carbohydrates in the stems of rice during grain filling.

### 3.2. Sink Strength Regulates the Remobilization of Carbon Reserves in Stems during Grain Filling

During the period of grain filling, the photo-assimilates can be accumulated in the storage organs of the stem, and then they can also be degraded and transported throughout the carbon reserve remobilization of the stems [13,27]. The SuSase, AGPase, and α-Amylase play an important role in the carbon reserve remobilization of the stem, but sufficiently biochemical data are needed to determine the change patterns of relative enzyme activity under the different sink strengths present [28]. After removing the spikelets, the efficiency of NSC transportation obviously decreased with the significant increase in sucrose content and starch content of the stem (Table 3 and Table 4). The SuSase (*OsSUS2*) acts out an important role in catalyzing sucrose cleavage into fructose and UDP-glucose, and AGPase (*OsAGPL1*) is essential for starch biosynthesis and strongly related to the sugar content [7,29,30]. The α-Amylase (*α-Amy3*) mainly stimulates the hydrolysis of stored starch with the carbohydrate starvation in rice [31]. The level of SuSase (*OsSUS2*) and AGPase (*OsAGPL1*) uniformly showed a relative increase, and the level of α-Amylase (*α-Amy3*) dramatically reduced in the stem under the spikelets removing treatment (Figure 3). In the same treatment, most of the data showed that the content of sucrose and starch in the stem of W1844 was significantly lower than that of CJ03 (Table 2). Interestingly, there are more obvious improvements in the grain filling of W1844 than that of CJ03 after removing spikelets (Table 1 and Table 2, Figure 1), and the efficiency of the NSCs’ transport is higher in W1844 than that in CJ03 (Table 3). These changes indicate that the sink strength of panicles might be related to regulation of the carbon reserve remobilization in the stems of rice during grain filling. Further analyses at different grain-filling stages reflected this phenomenon. During the early grain-filling period, most of the data in the levels of SuSase (*OsSUS2*), AGPase (*OsAGPL1*), and α-Amylase (*α-Amy3*) in W1844 were obviously lower than those of CJ03 at 8 DPA (Figure 3). Intriguingly, the level of SuSase (*OsSUS2*) was significantly higher in W1844 than that of CJ03 at 20 DPA, while the levels of AGPase (*OsAGPL1*) and α-Amylase (*α-Amy3*) in W1844 were not significantly higher than those of CJ03 (Figure 3). Based on our previous findings, the significant difference in the conversion of sucrose and starch in rice stems may be highly ascribed to the higher initiation of grain filling in W1844 than in CJ03, and the obvious difference in the grain growth of the IS between CJ03 and W1844 could also be a direct driver for the conversion of sucrose and starch in rice stems [4,5]. In addition, most of the values in CJ03 of the sucrose content and starch content stayed nearly the same from 8 DPA to 20 DPA, but a huge decrease occurred in W1844 during that period (Table 4). Collectively, these data demonstrate that sink strength plays a key role in carbon reserve remobilization. The sink strength of the rice panicle acts as an important driver in regulating the gene expressions and enzyme activities involved in the sucrose metabolism of rice stems among the different varieties during grain filling.

### 3.3. Sink Strength May Be Related to the Regulation of Sugar Signaling Involved in the Carbon Reserve Remobilization of Stems

Sucrose is unloaded by apoplastic transport in the stem, moving to the storage parenchyma cells and going through a complex process of carbon metabolism, which is an important step for sucrose transportation in the stem [32,33]. Consideration of the sugar-signaling network with regards to T6P and the SnRK1 pathway is essential for the regulation of carbon metabolism, but this has not been clear in the relationship among the sugar-signaling network of the stem, carbon metabolism of the stem, and sink strength in rice. After removing spikelets, the T6P content of the stem was significantly improved within the upregulated expression of *OsTPS8*, and most of the data showed obvious downregulation in gene expression of *OSK1*, *OSK24*, and *OsTPP1* (Figure 4 and Figure 5). Notably, the sucrose content of the stem was largely increased with great changes in the level of carbon metabolism after removing spikelets (Table 4, Figure 3). There are some reports of upregulation in the T6P pathway, increasing expression regarding TPS (*OsTPS8*) and TPP (*OsTPP1*) to regulate the growth process of the plant [21,34]. Moreover, T6P can affect its own downstream genes, partly overlapped by the downstream genes of SnRK1, by which it is found that SnRK1 can activate TPP transcription [35,36]. The genes of *OSK1* and *OSK24* are consistent in the SnRK1 family, which is correlated with carbohydrate metabolism [37]. These phenomena led to a query: what is the relationship between sink strength and the sugar-signaling pathway in the stem. In the same treatment, there was less T6P content and expression of *OsTPS8* in W1844 than in CJ03 at 8 DPA (Figure 4). Most of the data related to the sucrose content, the starch content, and the level of carbon metabolism in the stem were lower in W1844 than in CJ03 (Table 4, Figure 3). Moreover, the SnRK1 plays a central role in regulating the level of AGPase (*OsAGPL1*) and activating the level of α-Amylase (*α-Amy3*), which is strongly related to the response of sugar starvation in plants [38,39,40,41]. Hence, sink strength might be related to the regulation of sugar signaling involved in carbon metabolism of the stem, but the complex mechanism of the sugar-signaling network still needs to be explored. The T6P content, expression of *OsTPS8*, and level of SuSase (*OsSuS2*) were significantly higher in W1844 than in CJ03 at 20 DPA. However, there was no significant difference that existed in AGPase (*OsAGPL1*) and α-Amylase (*α-Amy3*), and expression of *OSK1*, *OSK24*, and *OsTPP1* between CJ03 and W1844 (Figure 3, Figure 4 and Figure 5). This phenomenon might be related to the high transport efficiency of soluble carbohydrates in W1844 during grain filling, which was proposed previously [4,5]. According to all the data and previous researches, the changes in the sugar-signaling network in stems may be compatible with a “feast-or-famine” model to adjust source–sink relations in rice [4,42,43,44]. Thus, the sink strength of the panicle could be a main regulator for the sugar-signaling pathway to drive carbon reserve remobilization in rice stems during grain filling.

## 4. Materials and Methods

### 4.1. Plant Material and Growth Condition

Rice (*Oryza sativa* L.) cultivars, CJ03 and W1844, were used to conduct field experiments (Danyang Experimental Base of Nanjing Agricultural University, Jiangsu Province, China). CJ03 and W1844 are homozygous large panicle japonica rice cultivars from the State Key Laboratory of Rice Genetics and Germplasm Innovation, Nanjing Agricultural University, which is sensitive to source–sink regulation with differential biomass production and yield [7]. For analyzing the relationship between grain filling and transportation of NSC in the stem, the field experiment was conducted at a hill spacing of 13.3 cm × 30 cm with two seedlings per hill in the rice grown seasons of 2019 and 2020. The plot’s size was 7 m × 10 m. The field experiment had a randomized complete block design with three replicates for each treatment. The soil was clay loam with 280 kg ha^−1^ nitrogen application (base fertilizer: panicle fertilizer = 5:5). The base fertilizer was applied before transplanting, and the panicle fertilizer was applied when the leaf-age remainder was 3.5.

Seeds were sown in the nursery beds on 21 May 2019 and 23 May 2020. The heading date (50% of plants) of CJ03 and W1844 was from 29–31 August 2019 and from 1–3 September 2020. We harvested from 2–4 November 2019 and from 3–5 November 2020. All agronomic management practices (e.g., cultivation and management measures) followed local recommendations.

### 4.2. Experimental Design

About 1400 panicles with similar growth patterns that headed on the same day were chosen and tagged. The flowering date of each position spikelet on the tagged panicles was recorded. On the flowering date of CJ03 and W1844, most labeled panicles were withdrawn from the flag leaf sheath completely, and spikelet-thinning treatment was performed according to the following protocol. The experiments included the control group with no spikelet thinning (T0), and the upper 2/3 followers removed group (T1) when the IS in the lower part of the panicles was flowering. The spikelet thinning way was the same as our previous study [4]. The superior spikelets (SS) were the grains on the three primary branches of the upper part of the panicle, while inferior spikelets (IS) were the grains on the three second branches in the lower part (Appendix A). The difference in flowering date between SS and IS was almost 4–5 days in the same panicle.

### 4.3. Sampling and Measurement

#### 4.3.1. Agronomic Analysis of Panicle

The panicles were harvested in 2019 and 2020 when approximately 85% of grains became yellow. The panicles were counted, and then dried at 80 °C for at least a week. The dry weights were determined to calculate 1000-grain weight and the number of filled spikelet. The seed setting rate was determined using the method by Kobata as follows [45]

Seed setting rate = plump grain number/total grain number
(1)

#### 4.3.2. Grain Weight and Grain Growth Rate

The grain-filling pattern of SS and IS in different treatments were measured in 2019 and 2020. Panicle samples were separated at 4, 8, 20, 40 DPA (days post anthesis) and maturity. In 2019 and 2020, the panicles of T0 treatment were separated into two parts: (1) spikelet on the three primary branches of the upper part of the panicle (SS), (2) spikelet on the three second-branches in the lower part of panicles (IS), and the panicles of the T1 group were separated into one group: (1) spikelet on the three second-branches in the lower part of panicles (IS). The SS and IS were dried in the oven at 105 °C for 0.5 h, and then dried at 80 °C for about one week, and then the grains of all groups were weighed and dehulled to determine the dry weight (DW). Grain-filling processes were fit to Richards’s growth equation [46].
(2)$$W=\frac{A}{{\left(1+Be^{-kt}\right)}^{1/N}}$$

*W* represents the grain weight (mg), *A* is the final grain weight (mg), *t* is the time after anthesis (days), and *B*, *k*, and *N* are coefficients established from the regression of the equation.

#### 4.3.3. Dry Matter Analysis

All the whole stems were sampled at 4, 8, 20, and 40 DPA. Among them, three averaged hills were separated into two parts: (1) panicles, (2) culms, and leaf sheath (stem). All samples were dried in the oven at 105 °C for 0.5 h, and then dried at 80 °C for about one week. The samples of all groups were weighed and dehulled to determine the dry weight (DW). The ground sample powder was stored in plastic bags at room temperature for chemical analysis. The remobilization rate of stem reserves, were calculated by DM as follows

ΔW (8–20 DPA) = stem dry matter (20 DPA) − stem dry matter (8 DPA)
(3)

Remobilization rate (%) = 100 × (W8 – W20)/W8
(4)

W8 is the peak of stem DM weight at 8 DPA, W20 is the lowest value at 20 DPA, depending on cultivars and treatments. For example, in the field experiment of 2019, stem DM was lowest at 20 DPA for the T0 group of W1844.

#### 4.3.4. Analysis of NSCs (Sucrose and Starch)

The NSC concentration in the whole stem was determined through a gravimetric method during the grain-filling stage (at 4, 8, 20, and 40 DPA) [47]. Furthermore, the NSCs’ content was calculated as follows

Stem NSC = stem NSC concentration (%) × stem weight/100
(5)

ΔNSC (8–20 DPA) = stem NSC (20 DPA) – stem NSC (8 DPA)
(6)

The sucrose content in the stem was calculated as mg/g fresh weight [48]. The sample was weighed about 100 mg powder per replicants and then extracted with 8 mL 80% aqueous ethanol at 80 °C for 0.5 h. After cooling for 10 min, the sample was centrifuged at 5000 rpm for 15 min. The sucrose extraction was repeated three times, and all the supernatants were combined and diluted to 50 mL. Moreover, activated carbon was then added to decolorize and purify. About 100 μL of 2 mol L^−1^ NaOH solution was added into 1.0 mL of the extract solution and boiled at 100 °C about 0.5 h before adding 3.0 mL of 10 mol L^−1^ HCl and 1.0 mL of 0.1% resorcinol. The reaction mixture was incubated at 80 °C for 10 min and determined at 480 nm.

The starch content in the stem was calculated as mg/g fresh weight [49]. The sample was weighed about 100 mg of powder per replicants and then added into the tubes with 2 mL of distilled water before boiling at 100 °C for 20 min. Two mL of 9.2 mol·L^−1^ HClO_4_ was added to the cooled tube and then vibrated for 10 min. All the samples were centrifuged at 5000 rpm for 15 min, and the supernatant collected into a 50 mL volumetric flask. This process was repeated three times. Then, all the supernatants were combined and diluted to 50 mL. About 0.1 mL of the extract and 4 mL of 0.2% anthrone were added into a new 15 mL centrifuge tube, which was placed into an 80 °C water bath for 15 min. The colorimetric determination was measured at OD 620 nm.

#### 4.3.5. Analysis of Enzyme Activities in Stem

All the whole stems were sampled at 8 DPA and 20 DPA after flowering, and frozen by liquid nitrogen for at least 1 min, and then stored at −80 °C for enzyme assays.

SuSase activity and AGPase activities were measured according to the method of Nakamura [50]. The stems were ground in a medium containing 50 mM Hepes-NaOH (pH 7.5), 10 mM MgCl_2_, 1 mM EDTA, 5 mM dithiothreitol (DTT), 1 mM phenazine methosulfate (PMSF), 1 mM benzamidine, and 3% (*w*/*v*) polyvinylpolypyrrolidone (PVPP). To obtain supernatant for assay of activities, the mixture was centrifuged at 4 °C and 12,000 rpm for 20 min. The dehulled and homogenized samples were stored at 0 °C. The homogenate was centrifuged at 15,000× *g* for 15 min at 0 °C after being filtered through four layers of cheesecloth, and then the supernatant was used for the enzyme assay, respectively.

For α-amylase activity, the stems were ground using 100 mM phosphate buffer (pH 6.5) and centrifuged at 4 °C and 12,000 rpm for 20 min. The supernatant was used for the assay of α-amylase activities [51]. About 0.1 mL of enzyme preparation was added to 0.6 mL of 12.7 mM calcium acetate (pH 6.0) and 0.3 mL of 0.067% soluble starch solution. To inactivate β-amylase, the assay mixture was incubated at 70 °C for 15 min. Then the assay mixture was incubated at 37 °C for 20 min and 0.1% I_2_–1% KI added to measure at 610 nm. For the inactive enzyme, the control group was heated in boiling water for 30 s. The α-amylase activity was calculated according to the control group and treatment group.

#### 4.3.6. Analysis of T6P Content

All the whole stems were sampled at 8 DPA and 20 DPA after flowering and frozen in liquid nitrogen for 1 min before storing at −80 °C for Trehalose-6-phosphate (T6P) assay. The samples were analyzed by Plant trehalose-6-phosphate synthase ELISA Kit (Shanghai Jianglai Biotech, Shanghai, China). The double antibody sandwich method was used for assay of T6P level. The T6P of the purified plant was used to capture the antibody and coat the microplate to make a solid-phase antibody. T6P was added into the coated microplate in turn and then combined with HRP-labeled detection antibody to form antibody–antigen enzyme-labeled antibody complex. TMB was added to develop the color after thorough washing. TMB is transformed into blue under the catalysis of the HRP enzyme and yellow under the action of acid. The supernatant was measured at 450 nm, and the content of T6P was calculated by standard curve.

#### 4.3.7. Analysis of Relative Genes Expression

All the whole stems were sampled at 8 DPA and 20 DPA, then frozen in liquid nitrogen for 1 min, before storing at −80 °C for RNA extraction. Gene transcription levels of these genes were analyzed through RNA extraction, cDNA synthesis, and quantitative real-time polymerase chain reaction (qRT-PCR). Plant RNA Kit (Omega Biotek, Inc., Norcross, GA, USA) was used to isolate the total RNA from the stems, and then the total RNA was reverse-transcribed into the first-strand cDNA with the Prime-Script-TM RT Reagent Kit (Takara, Kyoto, Japan), oligo-dT. qRT-PCR was performed using an ABI 7300 sequencer and SYBR Premix Ex Taq-TM (Takara, Kyoto, Japan) according to the manufacturer’s protocol. All experiments were conducted with three samples taken at each time point. The primers of this research were included in Appendix A.

### 4.4. Statistical Analysis

Statistical analyses were used by Student’s t-test over two years, according to similar performances in 2019 and 2020. The variance analysis was conducted using Duncan’s test, with statistical significance accepted at *p* < 0.05. Statistical analyses were performed using SPSS Statistics (SPSS Inc., Chicago, IL, USA). Illustrations were drawn in Microsoft Excel (Microsoft, Redmond, DC, USA) and Adobe Photoshop (Adobe, San Jose, CA, USA).

## 5. Conclusions

Our comprehensive analysis of the relationship between sink strength and carbon reserve remobilization of stems in CJ03 and W1844, within a two-year field experiment, shows that sink strength is responsible for regulating remobilization of NSCs by activating sugar signaling in rice stems during the grain filling. Compared with CJ03, WI844 is associated with the higher sink activity. Furthermore, the T0 group (without removing spikelet) has a larger sink size than the counterpart (T1). The high sink strength, involved by good sink activity and big sink size, substantially works to increase the remobilization of the carbon reserve by regulating the conversion of sucrose and starch in the rice stem. We find that the stronger sink strength may result in activating the sugar signaling of the T6P-SnRK1 pathway to upregulate the conversion of sucrose and starch in the rice stem. The comprehensive analysis of sink strength, the transport efficiency of carbohydrates in stems, the process of carbon reserve remobilization in stems, and the regulation of the sugar-signaling pathway in stems found new insights into novel roles in the regulation of the carbohydrate partition (Figure 6).

## Figures and Tables

**Figure 1 ijms-23-04864-f001:**
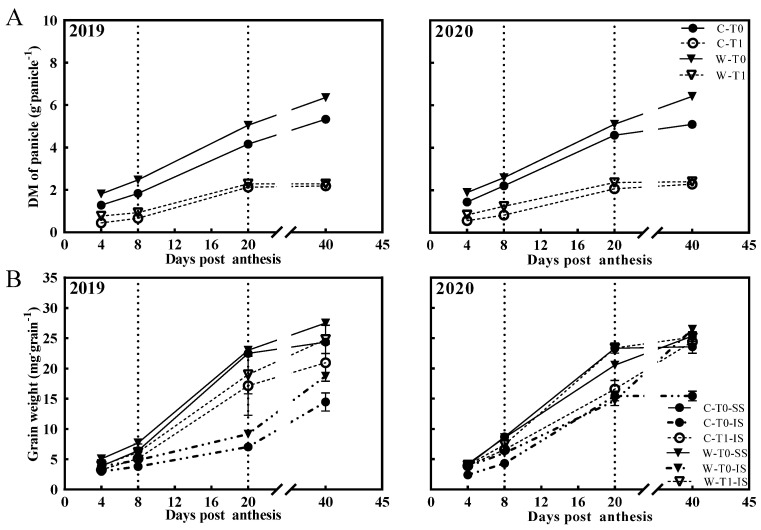
DM weight of panicles (**A**), SS and IS (**B**) during grain filling. C, CJ03; W, W1844; T0, control group; T1, top 2/3 of the spikelets were removed; SS, superior spikelets; IS, inferior spikelets; vertical bars represent mean values ± SE (*n* = 3).

**Figure 2 ijms-23-04864-f002:**
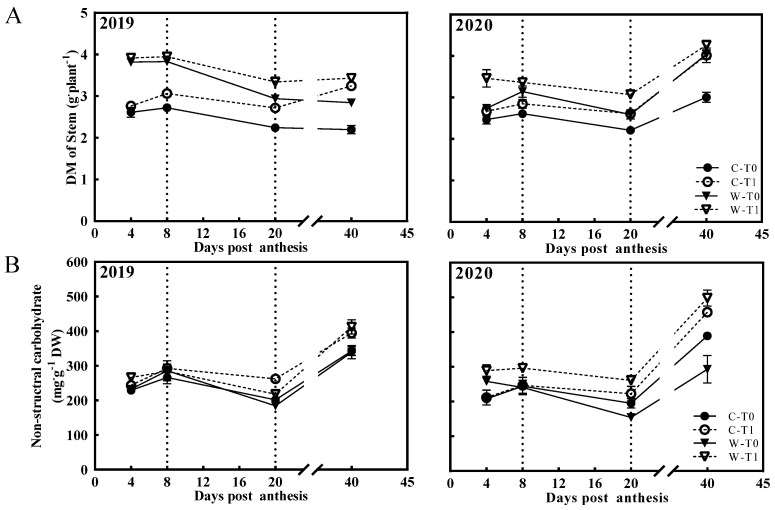
DM weight and NSC content of stems during grain filling. (**A**) DM weight of stem; (**B**) NSC content of stem; C, CJ03; W, W1844; T0, control group; T1, top 2/3 of the spikelets were removed. Vertical bars represent mean values ± SE (*n* = 3).

**Figure 3 ijms-23-04864-f003:**
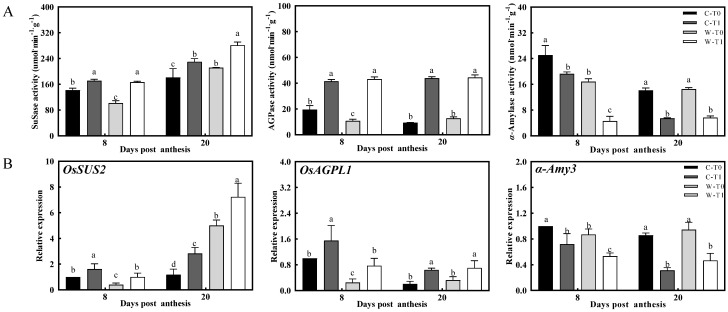
Activities of key enzymes and gene expression on carbohydrates’ metabolism of stems in 2020. (**A**) the activity of SuSase, AGPase, and α-Amylase in stem at 8 DPA and 20 DPA; (**B**) the expression of *OsSUS2*, *OsAGPL1*, and *α-Amy3* were validated by RT-PCR; C, CJ03; W, W1844; T0, control group without any treatment; T1, top 2/3 of the spikelets were removed; different letters from the same character under the same DPA indicate significant differences at the *p* = 0.05 level.

**Figure 4 ijms-23-04864-f004:**
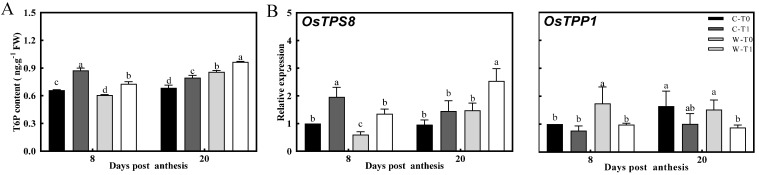
T6P content and gene expression with regards to the regulation of T6P level in the stems of 2020. (**A**) T6P content of stems during grain filling. (**B**) The expression of *OsTPS8* and *OsTPP1* in regulation of T6P level were validated by RT-PCR. C, CJ03; W, W1844; T0, control group; T0, control group without any treatment; T1, top 2/3 of the spikelets were removed; different letters from the same character under the same DPA indicate significant differences at the *p* = 0.05 level.

**Figure 5 ijms-23-04864-f005:**
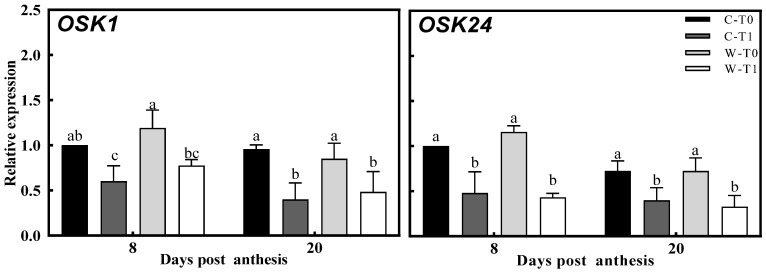
Gene expression with regards to the regulation of SnRK1 level in the stems of 2020. The expressions of *OSK1* and *OSK24* in the regulation of SnRK1 level were validated by RT-PCR. C, CJ03; W, W1844; T0, control group; T0, control group without any treatment; T1, top 2/3 of the spikelets were removed; different letters from the same character under the same DPA indicate significant differences at the *p* = 0.05 level.

**Figure 6 ijms-23-04864-f006:**
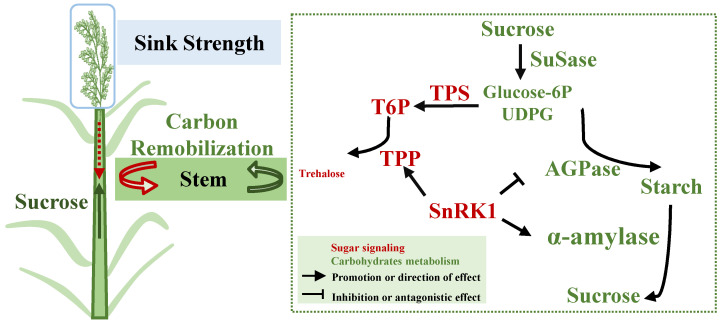
Model of metabolism and transportation of carbon in stem of rice during grain filling. Solid arrows direction effects; broken line indicates the indirection of effect; different colors indicate various metabolism.

**Table 1 ijms-23-04864-t001:** Agronomic traits of japonica rice materials at maturity.

Year	Variety	Treatment	Spikelets per Panicle	1000-Grain Weight (g)	Seed Setting Rate (%)
2019	CJ03	T0	267.67 a	21.87 c	87.67 c
		T1	91.17 b	20.10 d	93.00 b
	W1844	T0	275.67 a	22.91 b	84.92 d
		T1	97.92 b	25.95 a	95.58 a
2020	CJ03	T0	259.75 a	22.38 c	92.25 b
		T1	77.50 c	20.97 d	94.17 a
	W1844	T0	273.67 a	24.35 b	85.17 c
		T1	92.58 b	25.19 a	94.67 a

T0, control group without any treatment; T1, top 2/3 of the spikelets were removed; different letters from the same character under the same year indicate significant differences at the *p* = 0.05 level.

**Table 2 ijms-23-04864-t002:** 1000-grain weight and seed setting rate of rice materials at maturity.

Year	Variety	Treatment	1000-Grain Weight (g)		Seed Setting Rate (%)	
			Superior	Inferior	Superior	Inferior
2019	CJ03	T0	24.53 c	18.58 e	91.87 a	85.03 c
		T1	--	20.42 d	--	87.89 b
	W1844	T0	26.37 a	20.65 d	87.74 b	84.49 c
		T1	--	25.33 b	--	93.16 a
2020	CJ03	T0	22.88 b	17.62 d	92.74 a	85.27 cd
		T1	--	19.76 c	--	89.54 b
	W1844	T0	23.16 ab	20.70 c	87.54 bc	83.81 d
		T1	--	24.05 a	--	93.99 a

T0, control group without any treatment; T1, top 2/3 of the spikelets were removed; --, none data; different letters from the same character under the same year indicate significant differences at the *p* = 0.05 level.

**Table 3 ijms-23-04864-t003:** Changes in DM and NSCs’ accumulation from 8 DPA to 20 DPA.

Year	Variety	Treatments	ΔW (g·plant^−1^)	ΔNSCs (mg·g^−1^)	Remobilization of NSCs (%)
2019	CJ03	T0	−0.48 ab	−64.43 b	24.00 b
		T1	−0.34 a	−30.87 a	10.67 c
	W1844	T0	−0.89 c	−101.28 c	35.00 a
		T1	−0.60 b	−66.77 b	23.67 b
2020	CJ03	T0	−0.40 b	−48.66 a	20.33 b
		T1	−0.23 a	−23.95 a	10.00 c
	W1844	T0	−0.55 c	−86.72 b	35.67 a
		T1	−0.29 ab	−35.12 a	12.00 c

T0, control group without any treatment; T1, top 2/3 of the spikelets were removed; different letters from the same character under the same year indicate significant differences at the *p* = 0.05 level.

**Table 4 ijms-23-04864-t004:** Carbohydrate content of stem during grain filling in 2020.

Variety	Treatment	Sucrose Content (mg·g^−1^ FW)		Starch Content (mg·g^−1^ FW)	
		8 DPA	20 DPA	8 DPA	20 DPA
CJ03	T0	11.34 c	11.03 b	20.90 b	6.06 c
	T1	32.14 a	33.67 a	35.93 a	33.16 a
W1844	T0	11.06 c	5.72 c	6.53 c	4.54 c
	T1	24.34 b	8.62 bc	37.24 a	12.63 b

T0, control group without any treatment; T1, top 2/3 of the spikelets were removed; different letters from the same character under the same DPA indicate significant differences at the *p* = 0.05 level.

## Data Availability

Not applicable.

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
