# Peer review of "Sink Strength Promoting Remobilization of Non-Structural Carbohydrates by Activating Sugar Signaling in Rice Stem during Grain Filling"

_ijms, 2022, doi:10.3390/ijms23094864_

Round 1

Reviewer 1 Report

This manuscript is a new and very interesting research approach in which the influence of a changed grain sink size of rice is related to the carbohydrate metabolism in the stem. An excellent research approach. As known for nitrogen, a reduced sink size of grains is also influencing the remobilisation of carbohydrates from stems.

Some sections need significant improvement in terms of readability and comprehension (Results, part 2.3; 2.4; Discussion, part 3.1; captions Table and Figures). I strongly recommend the relationship shown in the Conclusion without showing a plant on the left of Figure 6; but only as a ‘model’ of compartments as shown on the right (the plant organs then have to be integrated).

Remarks/comments: see attachment

Author Response

Dear reviewers,

First of all, we would like to thank you for the time spent on reviewing our paper. Your comments and suggestions have definitively helped us to improve the work. We have studied very carefully the review reports, and tried our best to integrate all the recommendations and suggestions within this revised version.

In order for you to track the changes easily, we highlighted the main changes in red color in the revised version. Please see the attachment.

We also took this revision opportunity to double check the paper and corrected some minor errors.

We give below, point-by-point, a summary of the changes made in this revision. Our responses are indicated in bold while your comments are in normal format.

We hope that this revision gives satisfaction to your concerns and meets fully the high standard of the Journal.

Sincerely,

Prof. Ganghua LI

Nanjing Agricultural University

Reviewer #1:

This manuscript is a new and very interesting research approach in which the influence of a changed grain sink size of rice is related to the carbohydrate metabolism in the stem. An excellent research approach. As known for nitrogen, a reduced sink size of grains is also influencing the remobilisation of carbohydrates from stems.

Some sections need significant improvement in terms of readability and comprehension (Results, part 2.3; 2.4; Discussion, part 3.1; captions Table and Figures). I strongly recommend the relationship shown in the Conclusion without showing a plant on the left of Figure 6; but only as a ‘model’ of compartments as shown on the right (the plant organs then have to be integrated). Remarks/comments: see attachment.

Response: Thank you very much for taking your time to review this manuscript. I really appreciate all your comments and suggestions. Please find my revisions in the re-submitted files. Accordingly, we have uploaded a copy of the original manuscript with all the changes highlighted by using red words. Thank you for the kind comments again.

Reviewer 2 Report

Dear authors and editor,

the manuscript “Sink strength promoting remobilization of non-structural carbohydrates by activating sugar signaling in rice stem during grain-filling” aims at analysing the remobilization of non-structural carbohydrates in different rice varieties. The manuscript is well written, and both the methodology and the results are very clear. I like the Introduction, which provides a good background and is clear and straightforward. However, I suggest explicitly declaring your objectives at the end of this paragraph. The only gap I recognise in this manuscript is that it is not clearly explained what the main objective of the study was. Thus, the risk is that the manuscript may not be included in future research. I suggest improving this part.

Author Response

Response to the reviewers

Title: Sink strength promoting remobilization of non-structural carbohydrates by activating sugar signaling in rice stem during grain-filling

Authors: Zhengrong Jiang1, 2, Qiuli Chen1, Lin Chen1, Dun Liu1, Hongyi Yang1, Congshan Xu1, Jinzhi Hong1, Jiaqi Li1, Yanfeng Ding1, Soulaiman Sakr2, Zhenghui Liu1, Yu Jiang1, Ganghua Li1*

Manuscript no: ijms-1689208

First revision, 25 April 2022

Dear reviewers,

First of all, we would like to thank you for the time spent on reviewing our paper. Your comments and suggestions have definitively helped us to improve the work. We have studied very carefully the review reports, and tried our best to integrate all the recommendations and suggestions within this revised version.

In order for you to track the changes easily, we highlighted the main changes in red color in the revised version. Please see the attachment.

We also took this revision opportunity to double check the paper and corrected some minor errors.

We give below, point-by-point, a summary of the changes made in this revision. Our responses are indicated in bold while your comments are in normal format.

We hope that this revision gives satisfaction to your concerns and meets fully the high standard of the Journal.

Sincerely,

Prof. Ganghua LI

Nanjing Agricultural University

Reviewer #2:

Dear authors and editor,

the manuscript “Sink strength promoting remobilization of non-structural carbohydrates by activating sugar signaling in rice stem during grain-filling” aims at analysing the remobilization of non-structural carbohydrates in different rice varieties. The manuscript is well written, and both the methodology and the results are very clear. I like the Introduction, which provides a good background and is clear and straightforward. However, I suggest explicitly declaring your objectives at the end of this paragraph. The only gap I recognise in this manuscript is that it is not clearly explained what the main objective of the study was. Thus, the risk is that the manuscript may not be included in future research. I suggest improving this part.

Response: Thank you for your positive comments on our manuscript. We have polished the introduction to interpret the main objective more clearly (Please see lines 74-83). Accordingly, we have submitted our manuscript with tracked changes to highlight the revisions. We hope that it is now clearer. Thanks for your kind advice again.

Reviewer 3 Report

The manuscript titled “Sink strength promoting remobilization of nonstructural carbohydrates by activating sugar signal in rice stem during grain filling” is perfectly planned and conducted, and hence should be accepted for publication without any changes.

Author Response

Response to the reviewers

Title: Sink strength promoting remobilization of non-structural carbohydrates by activating sugar signaling in rice stem during grain-filling

Authors: Zhengrong Jiang1, 2, Qiuli Chen1, Lin Chen1, Dun Liu1, Hongyi Yang1, Congshan Xu1, Jinzhi Hong1, Jiaqi Li1, Yanfeng Ding1, Soulaiman Sakr2, Zhenghui Liu1, Yu Jiang1, Ganghua Li1*

Manuscript no: ijms-1689208

First revision, 25 April 2022

Dear reviewers,

First of all, we would like to thank you for the time spent on reviewing our paper. Your comments and suggestions have definitively helped us to improve the work. We have studied very carefully the review reports, and tried our best to integrate all the recommendations and suggestions within this revised version.

In order for you to track the changes easily, we highlighted the main changes in red color in the revised version. Please see the attachment.

We also took this revision opportunity to double check the paper and corrected some minor errors.

We give below, point-by-point, a summary of the changes made in this revision. Our responses are indicated in bold while your comments are in normal format.

We hope that this revision gives satisfaction to your concerns and meets fully the high standard of the Journal.

Sincerely,

Prof. Ganghua LI

Nanjing Agricultural University

Reviewer #3:

The manuscript titled “Sink strength promoting remobilization of nonstructural carbohydrates by activating sugar signal in rice stem during grain filling” is perfectly planned and conducted, and hence should be accepted for publication without any changes.

Response: Thank you for your work in process of reviewing the manuscript.

Round 2

Reviewer 1 Report

Excellent paper!

(2) Mistakes for correction:

line 225: 'promote' insted of 'promot'

line 331: 'spikelets' insted of 'spiekelets'